# Testicular hormones mediate robust sex differences in impulsive choice in rats

Caesar M Hernandez[1,2†], Caitlin Orsini[2‡], Alexa-Rae Wheeler[1], Tyler W Ten Eyck[1], Sara M Betzhold[2], Chase C Labiste[1], Noelle G Wright[1], Barry Setlow[2], Jennifer L Bizon[1]*

[1]Department of Neuroscience, University of Florida, Gainesville, United States; [2]Department of Psychiatry, University of Florida, Gainesville, United States

**Abstract** Impairments in choosing optimally between immediate and delayed rewards are associated with numerous psychiatric disorders. Such 'intertemporal' choice is influenced by genetic and experiential factors; however, the contributions of biological sex are understudied and data to date are largely inconclusive. Rats were used to determine how sex and gonadal hormones influence choices between small, immediate and large, delayed rewards. Females showed markedly greater preference than males for small, immediate over large, delayed rewards (greater impulsive choice). This difference was neither due to differences in food motivation or reward magnitude perception, nor was it affected by estrous cycle. Ovariectomies did not affect choice in females, whereas orchiectomies increased impulsive choice in males. These data show that male rats exhibit less impulsive choice than females and that this difference is at least partly maintained by testicular hormones. These differences in impulsive choice could be linked to gender differences across multiple neuropsychiatric conditions.

*For correspondence: bizonj@ufl.edu

Present address: †Department of Cell, Developmental and Integrative Biology, The University of Alabama at Birmingham, Birmingham, United States; ‡Department of Psychology, The University of Texas at Austin, Austin, United States

Competing interests: The authors declare that no competing interests exist.

## Introduction

Pronounced sex differences are evident in a range of neuropsychiatric disorders, but the underlying biological and behavioral phenotypes that contribute to such differences remain poorly understood. One aspect of behavior that associates with many neuropsychiatric conditions is intertemporal decision making (*Heerey et al., 2007*; *Bickel et al., 2012*; *Bickel et al., 2014*; *Steinglass et al., 2012*; *Wilbertz et al., 2013*; *Hoptman, 2015*; *Steward et al., 2017*), which involves choices between small rewards delivered immediately and larger rewards delivered in the future. Individuals choose large over small rewards in the absence of delays but will discount the value of the large rewards as a function of the delay before their delivery. Variation in intertemporal choice predicts a variety of life outcomes, with extreme biases in either direction associated with neuropsychiatric disease. For example, an inability to forgo immediate rewards in favor of larger rewards delivered in the future, or greater 'impulsive choice', associates with attention deficit hyperactivity disorder (*Wilbertz et al., 2013*) and substance use disorders (*Bickel et al., 2012*; *Bickel et al., 2014*). In contrast, a hallmark feature of the eating disorder anorexia nervosa is a maladaptive preference for delayed gratification (*Steinglass et al., 2012*; *Steward et al., 2017*). Notably, the prevalence of these conditions is sex-biased, with males exhibiting higher rates of 'impulsive' disorders (e.g., substance use, attention deficit hyperactivity disorder) and females exhibiting disproportionate rates of anorexia nervosa (*Rolls et al., 1991*; *Becker et al., 2017*).

Despite such associations, there remains a surprising lack of consensus regarding how biological sex influences impulsive choice. Some studies in humans report that males show greater impulsive choice than females (*Kirby and Maraković, 1996*) whereas others report the opposite (*Logue and Anderson, 2001*; *Reynolds et al., 2006*; *Smith and Hantula, 2008*; *Beck and Triplett, 2009*) or a lack of gender differences altogether (*Mischel and Underwood, 1974*; *Kirby and Maraković, 1995*;

*Silverman, 2003*; *Bembenutty, 2007*). Studies in rodents have been equally inconclusive, with findings ranging from no sex differences (*Perry et al., 2008*; *Eubig et al., 2014*; *Sackett et al., 2019*) to males being more impulsive than females (*Panfil et al., 2020*), to females being more impulsive than males (*Van Haaren et al., 1988*; *Perry et al., 2007*; *Weafer and de Wit, 2014*).

Even fewer studies have addressed the contributions of gonadal hormones to impulsive choice. Testosterone and estrogen are the primary hormonal drivers of developmental traits and characteristics in males and females, respectively (*Wieland et al., 1971*; *Wilson et al., 1981*; *Rogol et al., 2002*). Circulating levels of these hormones both during brain development and in fully mature adults can influence a variety of cognitive functions (*Korol, 2004*; *Spritzer et al., 2011*; *Frick et al., 2015*; *Wagner et al., 2018*; *Koss and Frick, 2019*). Studies show that circulating levels of these hormones associate with impulsive choice in both men and women (*Takahashi et al., 2006*; *Doi et al., 2015*). Similarly, impulsive choice can vary across the menstrual cycle in women (*Smith et al., 2014*; *Diekhof, 2015*; *Stanton, 2017*). Notably, however, while exogenous administration of supraphysiological levels of testosterone decreases impulsive choice in male rats (*Wood et al., 2013*), this effect is not observed in humans, at least at relatively low (replacement level) doses (*Ortner et al., 2013*). Moreover, it is unclear in either sex whether physiological levels of gonadal hormones are necessary for regulating impulsive choice.

Elucidating the role of sex and gonadal hormones in impulsive choice could offer important insights into both the etiology and treatment of neuropsychiatric diseases. The current study thus had two primary goals. The first was to thoroughly evaluate sex differences in naive adult male and female rats. The second was to determine whether gonadal hormones are necessary for sex-typical patterns of impulsive choice in each sex.

## Results

A timeline of the sequence of experimental procedures is shown in *Figure 1A*. The effects of sex on intertemporal choice were initially evaluated in naïve male and female rats. Subsequently, rats underwent sham or gonadectomy surgery followed by retesting.

### Effect of sex on intertemporal choice performance

Rats were food restricted and trained in operant chambers on a fixed delay intertemporal choice task in which they made discrete-trial choices between two levers, one that yielded a small food reward delivered immediately and another that yielded a large food reward delivered after a delay that ranged from 0 to 60 s (*Figure 1B*). Male and female rats did not differ in the numbers of free-choice trials completed in the task (percentage of trials omitted, $t_{(30)} = -1.39$, p=0.174). Moreover, in this experiment and all others reported below, all rats reduced their choice of the large reward as the delay to its delivery increased (main effects of delay, all ps < 0.005). Importantly, males and females significantly differed in their choices between the large, delayed and small, immediate rewards. As shown in *Figure 2A* (see *Figure 2—source data 1* for raw data), females made fewer choices of large, delayed rewards than males (two-factor ANOVA, main effect of sex: $F_{(1,30)}$=10.586, p=0.003), particularly when longer delays preceded delivery of the large reward (sex × delay interaction: $F_{(4,120)}$=8.572; p<0.001). Note, however, that choice performance in females was not modulated by estrous cycle phase (main effect of phase: $F_{(3,45)}$=1.618, p=0.221; phase × delay interaction: $F_{(12,180)}$=1.765, p=0.168; *Figure 2B*, see *Figure 2—source data 2* for raw data).

Other performance measures were assessed to provide additional insight into male and female differences in choice behavior. Specifically, rats' latencies to press the two levers and to omit responding on the forced-choice trials that began each trial block were evaluated to provide a measure of motivation to obtain the small and large rewards associated with each lever (*Setlow et al., 2003*; *Mai et al., 2012*; *Hernandez et al., 2017*). A three-factor, repeated measures ANOVA (sex × delay × lever) was used to analyze the lever press latencies and indicated main effects of sex ($F_{(1,27)}$=10.047, p=0.004), delay ($F_{(4,108)}$=65.403, p<0.001), and lever ($F_{(1,27)}$=14.258, p=0.001). Moreover, as shown in *Figure 3A* (see *Figure 3—source data 1* for raw data), all interactions among these variables were significant (delay × sex: $F_{(4,108)}$=9.206, p<0.001; delay × lever: $F_{(4,108)}$=36.690, p<0.001; sex × lever: $F_{(1,27)}$=17.510, p<0.001; and sex × lever × delay: $F_{(4,108)}$=7.823, p<0.001). To further elucidate these interactions, two-factor, repeated-measures ANOVAs (lever × delay) were used to analyze lever press latencies in males and females separately. While analyses of both males

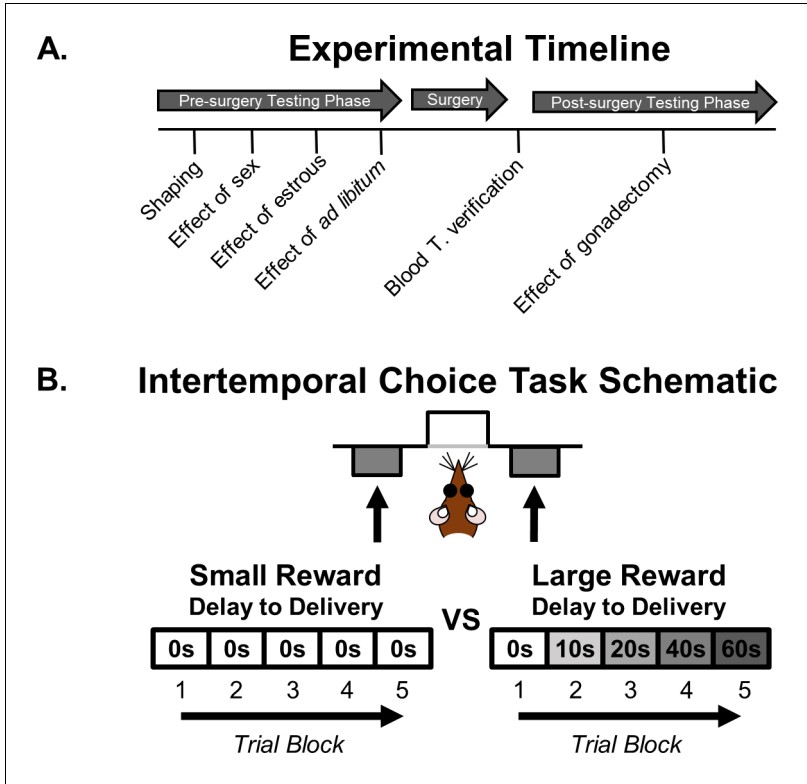

**Figure 1.** Intertemporal choice task and experiment timeline. (**A**) Timeline of experiments. (**B**) Schematic of the intertemporal choice task illustrating the choices and trial blocks across which the duration of the delay to the large reward increased. On each trial, rats were presented with two response levers that differed with respect to the magnitude and timing of associated reward delivery. A press on one lever delivered a small (one food pellet), immediate reward, whereas a press on the other lever delivered a large (four food pellets), delayed reward. Trials were presented in a blocked design, such that the delay to the large reward increased across successive blocks of trials in a session.

and females revealed main effects of delay (males: $F_{(4,60)}=17.453$, $p<0.001$; females: $F_{(4,40)}=28.762$, $p<0.001$) and delay × lever interactions (males: $F_{(4,60)}=8.603$, $p<0.001$; females: $F_{(4,48)}=27.137$, $p<0.001$), a main effect of lever was only present in females (males: $F_{(1,15)}=0.089$, $p=0.769$; females: $F_{(1,12)}=30.349$, $p<0.001$). Consistent with female preference for the small, immediate reward on free-choice trials, females showed much longer latencies to press the large compared to the small reward lever on forced-choice trials. In contrast, males showed comparable latencies for both levers. The percentage of omitted forced-choice trials was analyzed in a similar manner. As with the latency analysis, a three-factor repeated measures ANOVA (sex × lever × delay) conducted on the percentage of forced-choice trials omitted revealed main effects of sex ($F_{(1,30)}=16.363$, $p<0.001$), lever ($F_{(1,30)}=15.995$, $p<0.001$), and delay ($F_{(4,120)}=20.695$, $p<0.001$) as well as interactions across factors (sex × lever: $F_{(1,30)}=8.602$, $p=0.006$; sex × delay: $F_{(4,120)}=8.411$, $p<0.001$, lever × delay: $F_{(4,120)}=5.540$, $p<0.001$; sex × lever × delay: $F_{(4,120)}=5.156$, $p=0.001$; *Figure 3B*, see *Figure 3— source data 2* for raw data). Follow up analyses for males and females performed separately showed that females omitted more forced-choice trials on the large compared to small reward lever, particularly at the longest delays (main effect of lever: $F_{(1,15)}=13.988$, $p=0.002$; lever × delay: $F_{(4,60)}=7.526$, $p<0.001$). In contrast, no such differences were observed in males (main effect of lever: $F_{(1,15)}=2.015$, $p=0.176$; lever × delay: $F_{(4,60)}=0.214$, $p=0.888$). The longer response latencies and greater percentage of trial omissions associated with the large reward lever in females during forced-choice trials are consistent with females exhibiting greater impulsive choice and being less tolerant of delays than males. In addition, the fact that males and females did not differ in either response latencies or forced-choice trial omissions associated with the small reward lever indicates that females were as motivated as males to perform the task and seek rewards.

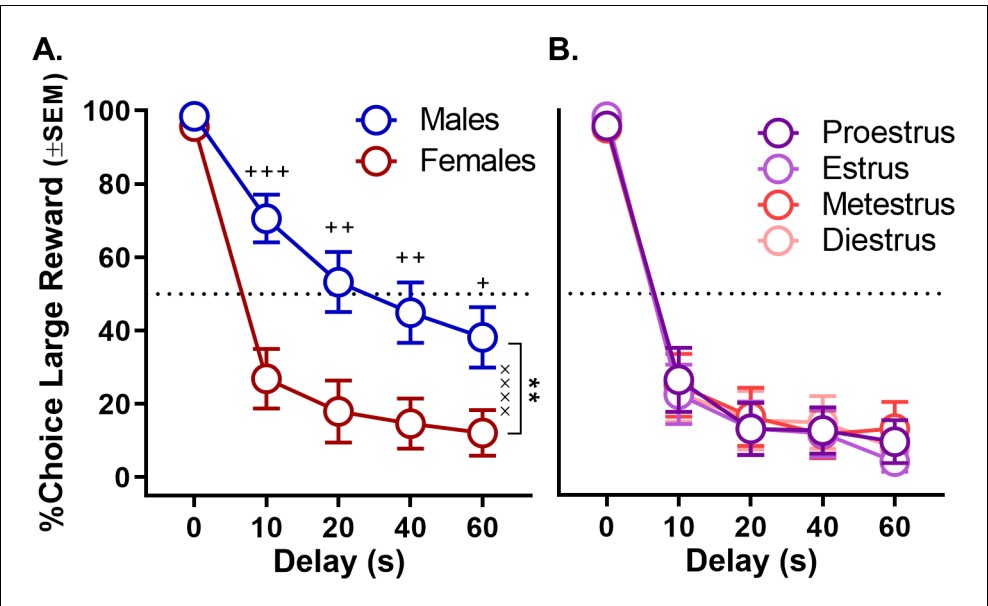

**Figure 2.** Effect of sex on impulsive choice. (**A**) Mean percent choice of the large reward in male and female rats. Female rats showed greater impulsive choice relative to males. (**B**) Impulsive choice in females did not fluctuate across the estrous cycle. In all panels, error bars represent the standard error of the mean (SEM). **p<0.01, main effect of sex; ××××p < 0.001, sex × delay interaction; +p<0.05, ++p<0.01, +++p<0.001, post-hoc t-tests sex difference at each delay. Raw data for these graphs are provided in *Figure 2—source data 1* and *2*.

The online version of this article includes the following source data for figure 2:

**Source data 1.** *Figure 2A* Sex difference in intertemporal choice.
**Source data 2.** *Figure 2B* Effect of estrous cycle phase on intertemporal choice.

## Effect of food motivational state on intertemporal choice performance in males and females

Adult male rats weigh substantially more than females and have greater ad libitum food intake (*Klump et al., 2013*). To determine whether potential sex differences in hunger level produced by the food restriction regimen might influence male and female differences in impulsive choice, rats were tested under two different satiation conditions. Specifically, in two separate experiments, rats were tested in the intertemporal choice task following 1 hr and 24 hr ad libitum access to their home cage chow. The 1 hr ad libitum schedule influenced overall motivation to perform the task, as evident by a main effect of feeding schedule on the percentage of free-choice trials omitted ($F_{(1,30)}$=9.115, p=0.005; *Figure 4C*, see *Figure 4—source data 3* for raw data). Importantly, however, this increase in free-choice trial omissions was present irrespective of sex (main effect of sex: $F_{(1,30)}$=0.294, p=0.592; schedule × sex: $F_{(1,30)}$=0.294, p=0.592). Note that two male and five female rats omitted entire blocks of free-choice trials, such that their data had to be excluded from analyses of choice behavior. A three-factor repeated measures ANOVA (sex × feeding schedule × delay) performed on choice data showed that the main effects of sex ($F_{(1,23)}$=26.753, p<0.001) and sex × delay interaction ($F_{(4,92)}$=11.227, p<0.001; *Figure 4A*, see *Figure 4—source data 1* for raw data) were still present following the 1 hr ad libitum feeding. In addition, 1 hr ad libitum feeding significantly decreased choice of large, delayed rewards (main effect of feeding schedule: $F_{(1,23)}$=5.861, p=0.024; *Figure 4B*, see *Figure 4—source data 2* for raw data), in a delay-independent manner (feeding schedule × delay: $F_{(4,92)}$=1.368, p=0.251). Most importantly, this effect of feeding schedule on choice behavior was independent of sex (schedule × sex: $F_{(1,23)}$=0.417, p=0.525; schedule × sex × delay: $F_{(4,92)}$=1.270, p=0.288).

The 24 hr ad libitum schedule also influenced overall motivation to perform the task as evident by a main effect of feeding schedule on the percentage of free-choice trials omitted (main effect of schedule: $F_{(1,30)}$=13.400, p=0.001; *Figure 4F*, see *Figure 4—source data 6* for raw data) that was

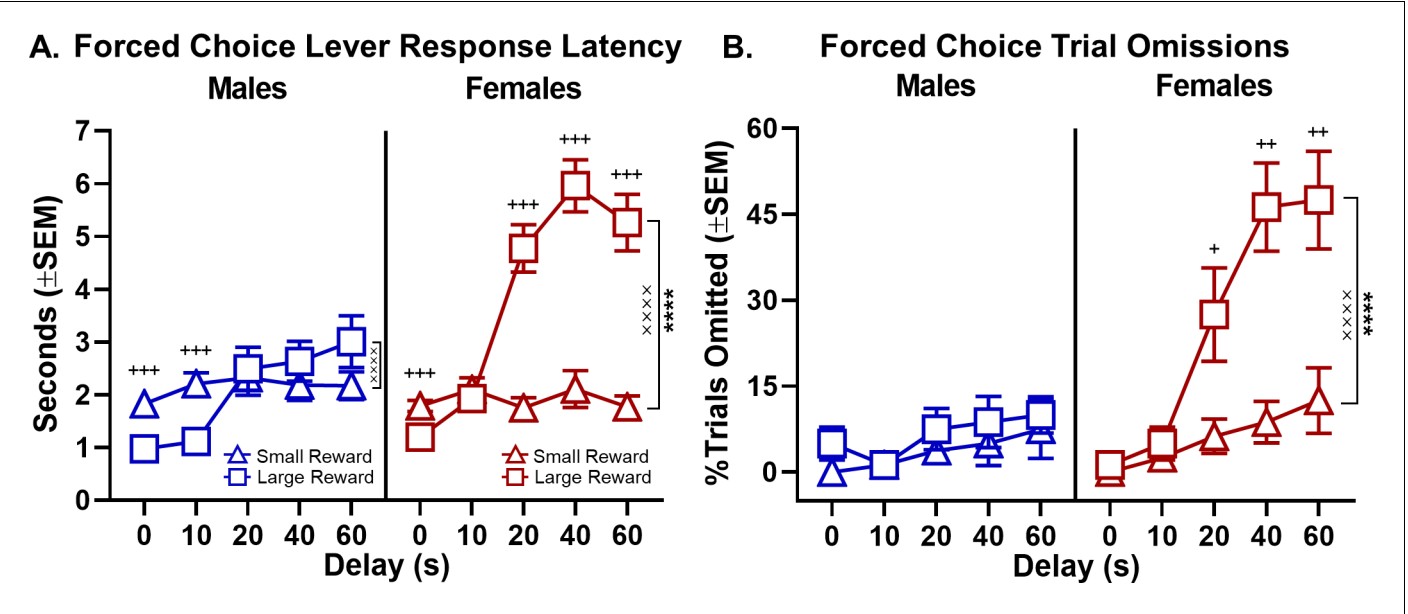

**Figure 3.** Effect of sex on forced-choice lever response latency and omissions. (**A**) As delays to large reward delivery increased, latency to press the large reward lever during forced-choice trials also increased in both sexes; however, this effect was more robust in females such that there was a main effect of lever in females but not in males. (**B**) While forced-choice trial omissions did not increase in males as a function of delay, females omitted significantly more forced-choice trials corresponding to the large reward lever at longer delays. In all panels, error bars represent standard error of the mean (SEM). ****p<0.001, main effect of lever; $^{\times\times\times\times}$p < 0.001, lever × delay interaction. $^{+}$p<0.05, $^{++}$p<0.01, $^{+++}$p<0.001, post-hoc t-tests lever latency/omission difference at each delay. Raw data for these graphs are provided in *Figure 3—source data 1* and *2*.

The online version of this article includes the following source data for figure 3:

**Source data 1.** *Figure 3A* Sex difference in forced choice lever response latencies.
**Source data 2.** *Figure 3B* Sex difference in forced choice trial omissions.

independent of sex (main effect of sex: $F_{(1,30)}$=2.297, p=0.140; schedule × sex: $F_{(1,30)}$=2.839, p=0.102). The 24 hr ad libitum feeding schedule caused three female rats to omit entire blocks of free-choice trials and therefore their data were excluded from analyses of choice behavior. A three-factor repeated measures ANOVA (sex × feeding schedule × delay) nevertheless demonstrated main effects of sex ($F_{(1,27)}$=13.414, p=0.001) as well as a sex × delay interaction ($F_{(4,108)}$=7.899, p<0.001; *Figure 4D*, see *Figure 4—source data 4* for raw data). Unlike the 1 hr ad libitum schedule, there was no decrease in choice of the large, delayed reward following the 24 hr ad libitum feeding schedule ($F_{(1,27)}$=2.302, p=0.141; schedule × sex: $F_{(1,27)}$=0.021, p=0.887; schedule × delay: $F_{(4,108)}$=0.179, p=0.949; schedule × sex × delay: $F_{(4,108)}$=1.433, p=0.228; *Figure 4E*, see *Figure 4—source data 5* for raw data). To summarize, given that robust differences in choice behavior between males and females were still evident following free access to food that was sufficient to reduce rats' motivation to perform the task, these data argue against the sex differences in intertemporal choice being attributable to differences in hunger or food motivation.

## Effects of gonadectomy on intertemporal choice

Prior to gonadectomy (GDX), males were divided into sham and orchiectomy groups, and females were divided into sham and ovariectomy groups. Baseline choice performance was matched across surgery groups separately for males and females. Due to post-operative complications, however, four female rats were euthanized and excluded for the remainder of the study. Pre-operative choice performance was equivalent between the male groups assigned to sham and orchiectomy conditions (no effect of group nor group × delay interactions: $Fs_{(1-4,14-56)}$=0.016–0.114, ps=0.900–0.977). Despite the attrition due to post-operative complications, pre-operative choice performance was equivalent between females assigned to sham and ovariectomy conditions (no effect of group nor group × delay interactions: $Fs_{(1-4,10-40)}$=2.146–2.257, ps=0.080–0.174).

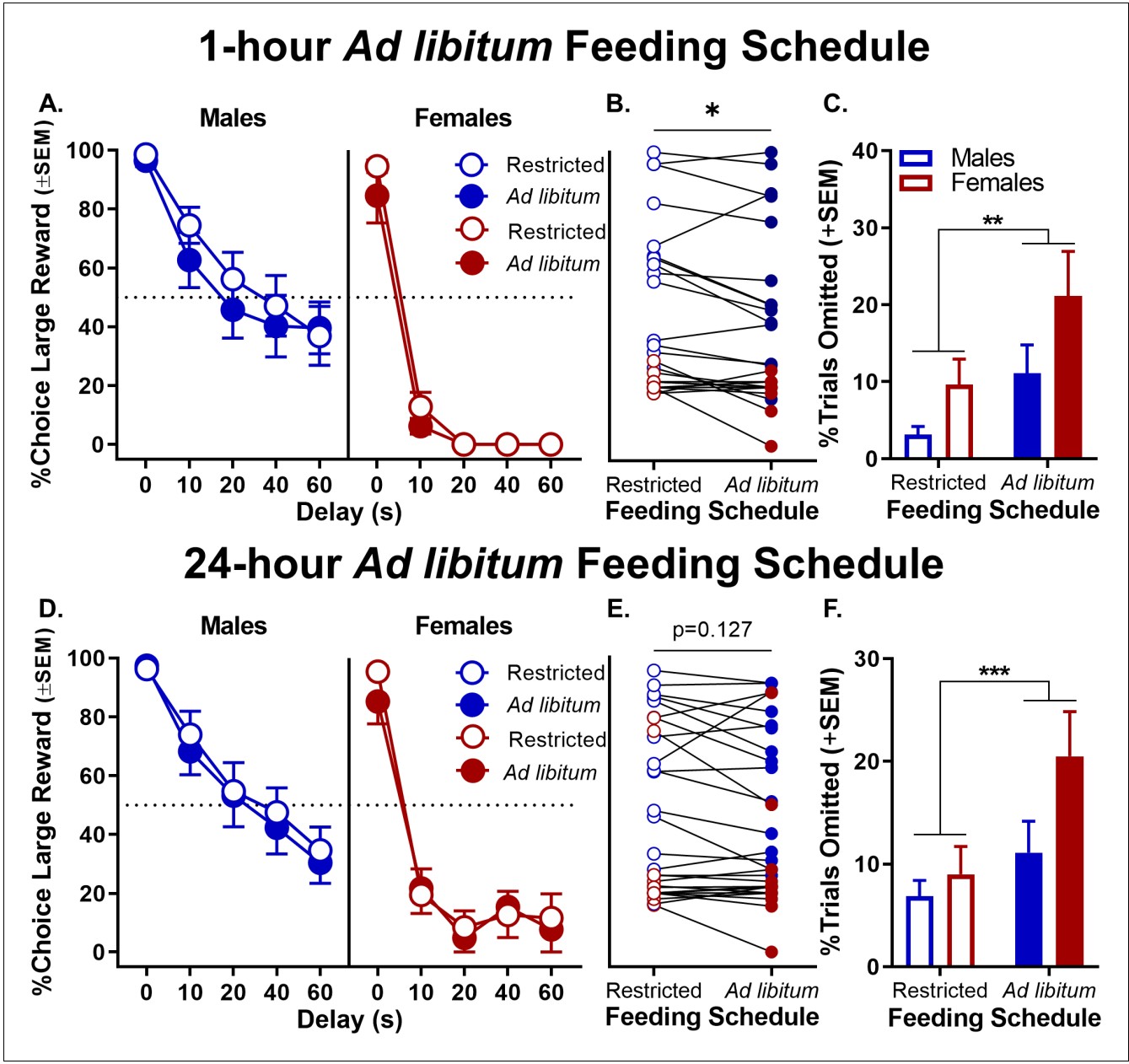

**Figure 4.** Effect of ad libitum feeding schedule on choice performance. (A) Relative to a food restriction feeding schedule, ad libitum feeding for 1 hr prior to testing shifted rats' preference towards small, immediate rewards, but did not differentially affect choice performance in males and females. (B) There was a main effect of ad libitum feeding relative to food-restricted feeding, averaged across all delays shown in A. Each rat's percentage choice of the large reward averaged across all delays is represented by an individual data point. (C) As expected, 1 hr of ad libitum feeding prior to testing increased free-choice trial omissions in all rats. (D-F) Similar methodology and data presentation as A-C except that data were collected after ad libitum feeding for 24 hr prior to testing. As with the 1 hr ad libitum schedule, the 24 hr ad libitum feeding did not differentially affect choice performance in males and females. In all panels, error bars represent standard error of the mean (SEM). *p<0.05, **p<0.01, ***p<0.001. Raw data for these graphs are provided in *Figure 4—source data 1–6*.

The online version of this article includes the following source data for figure 4:

**Source data 1.** *Figure 4A* Effect of 1-hour ad libitum feeding on intertemporal choice.
**Source data 2.** *Figure 4B* Effect of 1-hour ad libitum feeding on intertemporal choice (collapsed across delays).
**Source data 3.** *Figure 4C* Effect of 1-hour ad libitum feeding on free-choice trial omissions.
**Source data 4.** *Figure 4D* Effect of 24-hour ad libitum feeding on intertemporal choice.
**Source data 5.** *Figure 4E* Effect of 24-hour ad libitum feeding on intertemporal choice (collapsed across delays).
**Source data 6.** *Figure 4F* Effect of 24-hour ad libitum feeding on free-choice trial omissions.

Following ovariectomy surgeries, rats were confirmed as acyclic via vaginal lavage and cytology. A three-factor, repeated measures ANOVA (surgical group [ovariectomy vs. sham] × phase [pre- vs. post-surgery] × delay) was used to analyze the percentage of large reward choices and indicated no main effects or interactions involving ovariectomy (main effect of surgical group: $F_{(1,10)}=2.245$, p=0.165; main effect of phase: $F_{(1,10)}=0.003$, p=0.957; surgical group × phase: $F_{(1,10)}=0.001$, p=0.971; phase × delay: $F_{(4,40)}=1.557$, p=0.204; surgical group × delay: $F_{(4,40)}=1.655$, p=0.179; surgical group × phase × delay: $F_{(4,40)}=1.698$, p=0.170; *Figure 5A*, see *Figure 5—source data 1* for raw data). There were also no effects of phase on the percentage of trials omitted (main effect of phase: $F_{(1,10)}=2.390$, p=0.153; main effect of surgical group: $F_{(1,10)}=2.726$, p=0.130; surgical group × phase: $F_{(1,10)}=0.850$, p=0.378; *Figure 5B*, see *Figure 5—source data 2* for raw data). To address the possibility that the absence of effects of ovariectomy on choice performance was due to a floor effect (i.e., because rats already showed low levels of preference for the large, delayed reward), female rats were subsequently trained using a shorter set of delays to large reward delivery (0, 2, 4, 8, 16 s). One sham rat was excluded from this analysis due to omissions of complete blocks of trials, and one ovariectomy rat was excluded due to premature death. A two-factor, repeated measures ANOVA (surgical group × delay) revealed no effects of ovariectomy under the shorter delay set (main effect of surgical group: $F_{(1,8)}=0.006$, p=0.938; surgical group × delay: $F_{(4,32)}=0.407$, p=0.802), suggesting that the failure of ovariectomy to influence choice performance was not due insufficient parametric space in which to observe such effects (*Figure 5C*, see *Figure 5—source data 3* for raw data). Instead, these data are consistent with the interpretation that circulating ovarian hormones in adult females do not drive sex differences observed in intertemporal choice and are not critical for maintenance of the female-typical pattern of intertemporal choice behavior.

In males, orchiectomies were confirmed by measuring blood serum levels of testosterone using ELISA. Testosterone levels were detectable in sham but not orchiectomized rats ($t_{(14)} = 2.315$, p=0.036; sham: 1.892 ng/mL ±0.817 SEM; orchiectomized: 0 ng/mL ±0 SEM). Using a three-factor repeated measures ANOVA, the analysis of choice data (surgical group [orchiectomy vs.

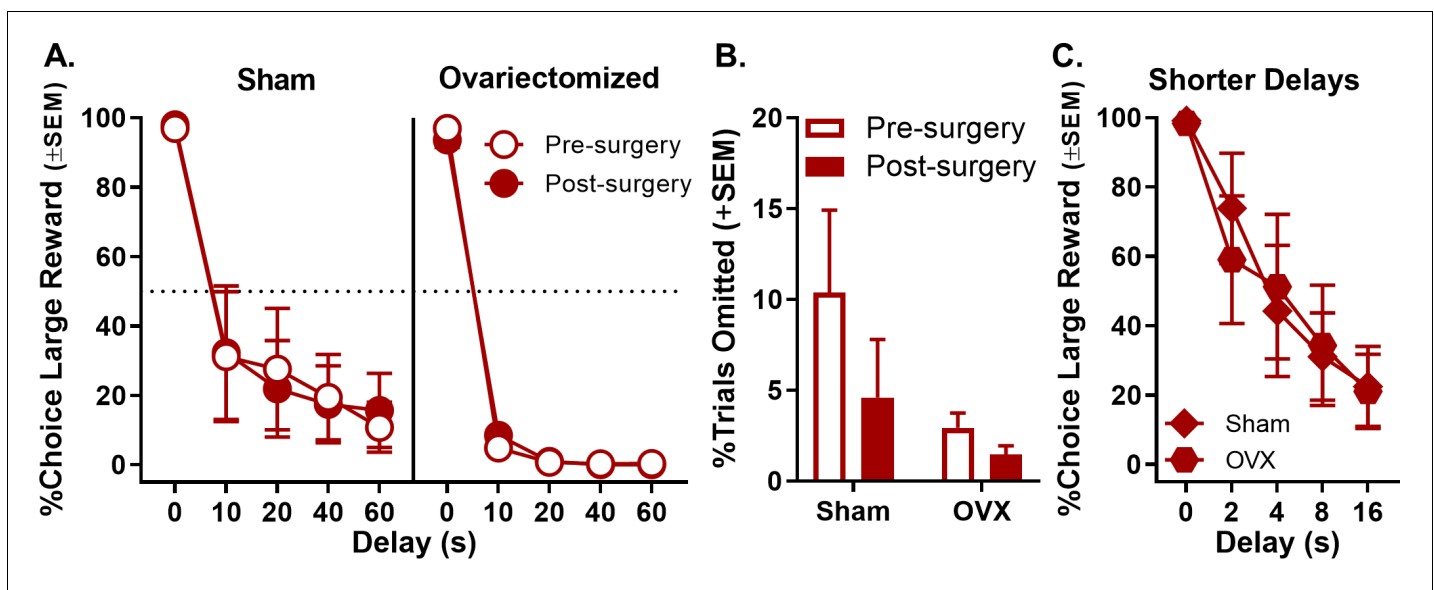

**Figure 5.** Effect of ovariectomy (OVX) on choice performance in female rats. (A) There were no significant changes in choice performance in either OVX or sham females after surgery. (B) Free-choice trial omissions in sham and ovariectomized female groups prior to and post surgeries. OVX had no effect on free-choice trial omissions. (C) Adjusting the intertemporal choice task to incorporate shorter delays did not reveal an effect of OVX. In all panels, error bars represent standard error of the mean (SEM). Raw data for these graphs are provided in *Figure 5—source data 1–3*.

The online version of this article includes the following source data for figure 5:

**Source data 1.** *Figure 5A* Effect of ovariectomy on intertemporal choice.
**Source data 2.** *Figure 5B* Effect of ovariectomy on free-choice trial omissions.
**Source data 3.** *Figure 5C* Effect of ovariectomy on intertemporal choice (shorter delays).

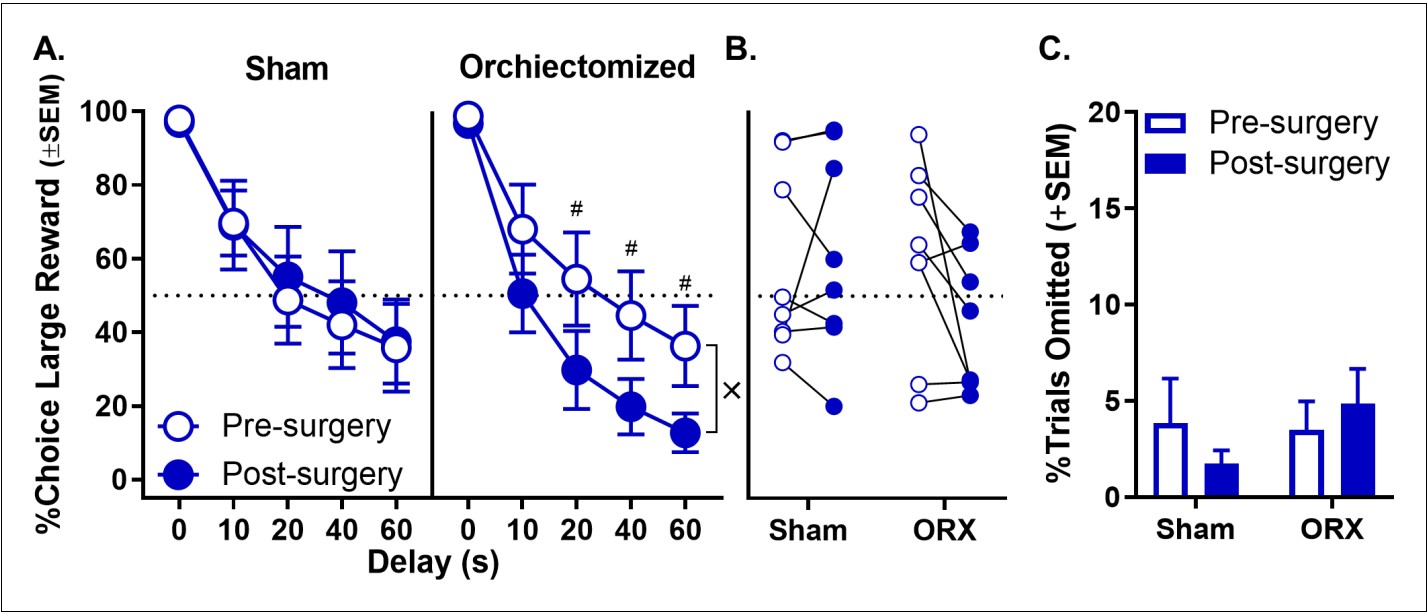

**Figure 6.** Effect of orchiectomy (ORX) on choice performance in male rats. (**A**) Choice performance in shams was unchanged post-surgery relative to pre-surgery; however, ORX caused an increase in impulsive choice post-surgery relative to pre-surgery. (**B**) Main effect of surgery in both sham and orchiectomized rats. Each rat's percent choice of the large reward (averaged across all delays) is represented by an individual data point. (**C**) Free-choice trial omissions in sham and orchiectomized male groups prior to and post surgeries. ORX had no effect on free-choice trial omissions. In all panels, error bars represent standard error of the mean (SEM). $^{\times}$p<0.05, group × phase × delay interaction. $^{\#}$p<0.07, post-hoc t-tests group difference at each delay. Raw data for these graphs are provided in *Figure 6—source data 1–3*.

The online version of this article includes the following source data for figure 6:

**Source data 1.** *Figure 6A* Effect of orchiectomy on intertemporal choice.
**Source data 2.** *Figure 6B* Effect of orchiectomy on intertemporal choice (collapsed across delays).
**Source data 3.** *Figure 6C* Effect of orchiectomy on free-choice trial omissions.

sham] × phase [pre- vs. post-surgery] × delay) revealed a trend toward a surgical group × phase interaction ($F_{(1,14)}$=3.750, p=0.073), and a significant surgical group × phase × delay interaction ($F_{(4,56)}$=2.569, p=0.048; *Figure 6A*, see *Figure 6—source data 1* for raw data), with no other effects reaching significance (main effect of surgical group: $F_{(1,14)}$=0.661, p=0.430; main effect of phase: $F_{(1,14)}$=2.153, p=0.164; phase × delay: $F_{(4,56)}$=0.951, p=0.429; surgical group × delay: $F_{(4,56)}$=0.560, p=0.692). *Figure 6B* shows that orchiectomized but not sham rats reduced their choice of the large, delayed reward post-surgery compared to pre-surgery conditions. To confirm this observation, additional two-factor, repeated measures ANOVAs were used to analyze choice data in orchiectomized and sham groups separately. These analyses indicated a trend toward a main effect of phase ($F_{(1,7)}$=4.829, p=0.064; *Figure 6B*, see *Figure 6—source data 2* for raw data) and a significant phase × delay interaction ($F_{(4,28)}$=3.029, p=0.034) in orchiectomized rats, such that choice of the large reward decreased post-surgery, particularly at the longest delays. In contrast, there was neither a main effect of phase nor a phase × delay interaction in sham rats (main effect of phase: $F_{(1,7)}$=0.138, p=0.722; phase × delay: $F_{(4,28)}$=0.407, p=0.687). Finally, orchiectomy did not influence the percentage of free-choice trials omitted (main effect of surgical group: $F_{(1,14)}$=0.467, p=0.505; main effect of phase: $F_{(1,14)}$=0.087, p=0.773; surgical group × phase: $F_{(1,14)}$=1.836, p=0.197; *Figure 6C*, see *Figure 6—source data 3* for raw data).

To further evaluate the effects of orchiectomy, additional analyses (of latencies and omissions on forced-choice trials) were conducted specifically in the orchiectomy group. A three-factor repeated measures ANOVA (phase × lever × delay) was used to analyze lever press latencies and revealed significant phase × lever ($F_{(1,7)}$=51.858, p<0.001), lever × delay ($F_{(4,28)}$=8.950, p=0.003), and phase × lever × delay ($F_{(4,28)}$=3.619, p=0.017) interactions (*Figure 7A*, see *Figure 7—source data 1* for raw data). To further elucidate these interactions, separate two-factor, repeated measures ANOVAs

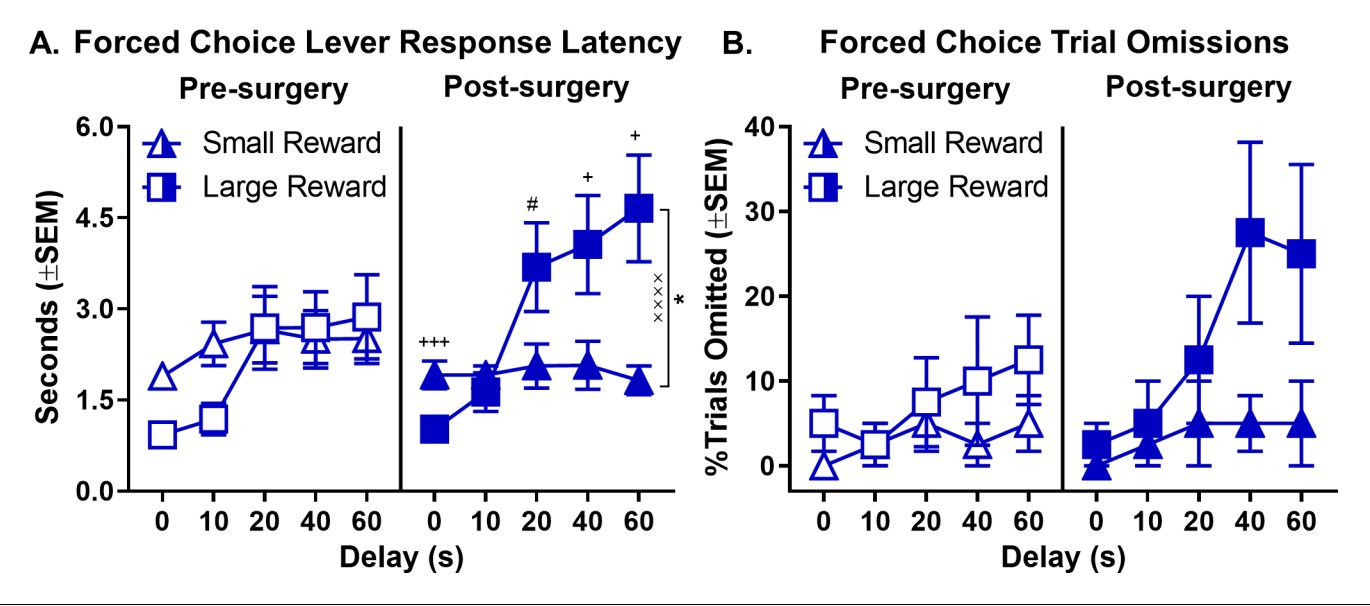

**Figure 7.** Effect of orchiectomies on forced-choice lever response latency and omissions. (A) As delays to large reward delivery increased, large reward lever response latency during forced-choice trials also increased in males in the orchiectomy group; however, the effect was more robust after orchiectomy, such that there was a main effect of lever that was absent pre-surgery. (B) While forced-choice trial omissions did not increase at longer delays prior to surgery in orchiectomized rats, after surgery, the rats numerically (though not significantly) omitted more large reward forced-choice trials at longer delays. In all panels, error bars represent standard error of the mean (SEM). *p<0.05, main effect of lever; ××××p < 0.001, lever × delay block interaction. #p<0.07, +p<0.05, +++p<0.001, post-hoc t-tests lever latency difference at each delay. Raw data for these graphs are provided in *Figure 7—source data 1* and *2*.

The online version of this article includes the following source data for figure 7:

**Source data 1.** *Figure 7A* Effect of orchiectomy on forced choice lever response latencies.
**Source data 2.** *Figure 7B* Effect of orchiectomy on forced choice trial omissions.

(lever × delay) were used to analyze lever press latencies pre- and post-surgery. While there were no differences in latencies to press levers pre-surgery (main effect of lever: $F_{(1,7)}$=0.340, p=0.578; lever × delay interaction: $F_{(4,28)}$=2.774, p=0.103), latencies to press the large reward lever were significantly greater than those for the small reward lever post-surgery, particularly at the longest delays (main effects of lever: $F_{(1,7)}$=5.911, p=0.045; lever × delay interaction: $F_{(4,28)}$=11.501, p<0.001). Orchiectomized males also showed a significant increase in the percentage of omitted forced-choice trials corresponding to the large reward at the longest delays (*Figure 7B*, see *Figure 7—source data 2* for raw data); the main effects of lever ($F_{(1,7)}$=5.804, p=0.047) and delay ($F_{(4,28)}$=2.982, p=0.036), however, were evident both pre- and post-surgery (main effect of phase: $F_{(1,7)}$=1.288, p=0.294; phase × lever: $F_{(1,7)}$=0.881, p=0.379; phase × delay: $F_{(4,28)}$=1.782, p=0.160; lever × delay: $F_{(4,28)}$=1.894, p=0.139; phase × lever × delay: $F_{(4,28)}$=0.645, p=0.635). Taken together, these data show that removal of testicular hormones shifts male rats' behavior toward a phenotype more similar to that of gonadally-intact females, such that both impulsive choice and delay intolerance are increased.

## Discussion

The current study used a rat model to explicitly evaluate the contributions of sex and gonadal hormones to intertemporal choice behavior. Relative to males, female rats were markedly more impulsive in their choices, preferring small, immediate over larger delayed rewards. This effect was not due to differences in reward magnitude discrimination, as both males and females consistently chose the large reward when there was no delay to its delivery. The effect was also not due to differences in task engagement (free-choice trial omissions) nor to differences in hunger-dependent states between males and females, as ad libitum feeding prior to testing did not alter the observed sex

differences. Instead, the current data are consistent with the interpretation that female rats are much less tolerant of delays than males. Importantly, while these sex-dependent effects were not influenced by removal of ovarian hormones, removal of testicular hormones resulted in greater impulsive choice in male rats, indicating a critical role for testicular but not ovarian hormones in maintenance of sex-typical patterns of impulsive choice.

## Females are more delay intolerant than males

One possible explanation for greater impulsive choice in females relative to males is a greater sensitivity to delay (or delay intolerance), as suggested by the analyses of the latencies to press the small, immediate vs. the large, delayed reward lever on forced-choice trials. Response latencies may provide insight into the affective and/or motivational processing underlying a response; longer latencies may indicate negative affect or reduced motivation for an action, whereas shorter response latencies may indicate positive affect or heightened motivation (*Crespi, 1942*; *Amsel, 1950*; *Setlow et al., 2003*; *Mai et al., 2012*; *Orsini et al., 2015*; *Shimp et al., 2015*; *Hernandez et al., 2017*). Consistent with this interpretation, both males and females in the current study had shorter latencies to respond for the large reward relative to the small reward under no delay conditions, and large reward response latencies in both sexes increased as delays to the large reward increased. Relative to males, however, females took significantly longer in responsding to the large reward in the presence of delays. Females also displayed an increase in the number of omitted large reward forced-choice trials in the presence of the longest delays, consistent with delay intolerance (i.e., at those delays, the large reward was so reduced in value as to not be worth pursuing at all). The current data indicate, however, that females were not less sensitive to reward than males. Indeed, females responded as rapidly as males in the absence of a delay. Hence, at least in the context of the food rewards and delays employed here, females and males are similarly sensitive to reward magnitude but females are substantially less tolerant of delays compared to males. Interestingly, these results parallel findings from across species that females are more sensitive to punishment than males (*Miettunen et al., 2007*; *Rood et al., 2009*; *Cross et al., 2011*; *Orsini et al., 2016*; *Chowdhury et al., 2019*; *Grissom and Reyes, 2019*; *Liley et al., 2019*). In this context, females may perceive the delays in the intertemporal choice task as a form of 'cost' that shares overlapping features with explicit punishment.

Delay intolerance in females as a primary driver of sex differences in intertemporal choice could account for some of the disparate findings in the rodent literature. Several prior studies failed to detect sex differences on intertemporal choice tasks (*Perry et al., 2008*; *Sackett et al., 2019*) whereas the results of other studies largely agree with the current findings (*Van Haaren et al., 1988*; *Perry et al., 2007*; *Koot et al., 2009*; *Eubig et al., 2014*; *Lukkes et al., 2016*). In particular, the relatively long delays employed in the present study (0–60 s) may have been optimal for detecting sex differences driven by delay intolerance. Indeed, several studies using shorter delays (0–40 s or 0–20 s) failed to detect sex differences (*Eubig et al., 2014*; *Sackett et al., 2019*).

In addition to the use of longer delays, other factors could have contributed to the robust sex difference observed in the present study, relative to previous findings of smaller or equivocal differences. Several prior studies found that females had numerically but not significantly greater impulsive choice compared to males (*Eubig et al., 2014*; *Lukkes et al., 2016*). These studies, however, employed relatively small group sizes (n = 6–9/sex) compared to the current study (n = 16/sex), which could account in part for their failure to detect significant sex effects. Another difference between the current and prior studies concerns the age at which the animals were tested. In the current study, rats began testing at 6 months of age, whereas rats in prior studies in which sex differences were not detected began testing at 3–4 months of age (*Perry et al., 2008*; *Sackett et al., 2019*). Impulsive choice in both sexes is reported to decrease from adolescence (younger than 2 months) to young adulthood (greater than 2 months; *Doremus-Fitzwater et al., 2012*) and in male rats it decreases further from 6 months to 24 months (*Simon et al., 2010*; *Hernandez et al., 2017*). To our knowledge, changes in impulsive choice over the course of young adult maturation have not been investigated. Thus, it is possible that sex differences in impulsive choice become more pronounced over the course of young adulthood. It is not likely that the design of the intertemporal choice task alone (fixed delays proceeding in an ascending order) accounts for the presence of robust sex differences, as previous studies that both have and have not detected sex differences have employed this design as well as adjusting delay designs (*Perry et al., 2007*; *Perry et al., 2008*;

*Sackett et al., 2019*; *Panfil et al., 2020*). Finally, it should be noted that previous studies have not evaluated sex differences in intertemporal choice in Fischer 344 × Brown Norway F1 hybrid rats, and thus it is possible that sex differences are particularly pronounced in this strain. Sex differences in many aspects of cognition (including other forms of cost-benefit decision making) have been observed in several rat strains (*van den Bos et al., 2012*; *Peak et al., 2015*; *Orsini et al., 2016*; *Blaes et al., 2019*; *Liley et al., 2019*), and several studies have shown differences in impulsive choice between rat strains (*Huskinson et al., 2012*; *Garcia and Kirkpatrick, 2013*). To our knowledge, however, no studies have directly compared sex differences in impulsive choice between strains, suggesting that this will be an important avenue of future research.

## Greater impulsive choice in females is not dependent on differential hunger/satiety

Prior studies have reported that the degree of food restriction can affect impulsive choice (*Bradshaw and Szabadi, 1992*; *Wogar et al., 1992*; *Ho et al., 1997*), such that greater restriction leads to lower levels of impulsive choice. Because the rats in the current study were food-restricted to 85% of their free-feeding body weights, one possible explanation for greater impulsive choice in females relative to males is that females experienced a subjectively different level of food restriction than males. Manipulation of restriction level through ad libitum feeding modestly decreased choices of large, delayed rewards, but this effect did not differ by sex. Additionally, although more females than males omitted entire blocks of free-choice trials under satiated conditions, there was a similar increase in the overall number of free-choice trial omissions in males and females, suggesting that both sexes experienced a comparable degree of satiation. As such, these findings argue against the interpretation that greater impulsive choice in females is due to a difference in hunger state.

## Testicular but not ovarian hormones are necessary for sex-specific patterns of impulsive choice

There is evidence that impulsive choice in women varies across the menstrual cycle, suggesting that preference for small, immediate vs. large, delayed rewards can be modulated by fluctuations in ovarian hormones within the physiological range (*Kaighobadi and Stevens, 2013*; *Smith et al., 2014*; *Diekhof, 2015*). To determine whether this variation is also evident in rats, impulsive choice was evaluated across phases of the estrous cycle. Although previous work shows that some aspects of cognition and behavior in rats can fluctuate across the estrous cycle (*Shansky et al., 2006*; *Tuscher et al., 2015*; *Becker and Koob, 2016*; *Blume et al., 2017*; *Pellman et al., 2017*; *Yagi et al., 2017*), there was no effect of estrous phase on impulsive choice in the current study. This finding is consistent with those from previous studies of sex differences in decision making involving risk of punishment or delayed punishment, in which estrous cycle does not modulate choice behavior (*Orsini et al., 2016*; *Liley et al., 2019*). Similarly, estrous phase does not influence rats' accuracy in a delayed response working memory task, performance on which is correlated with impulsive choice (*Shimp et al., 2015*; *Blaes et al., 2019*).

Removal of ovarian hormones through ovariectomy can alter some forms of learning and memory (*Bimonte-Nelson et al., 2003*; *Qu et al., 2013*; *Frick et al., 2015*; *Kim et al., 2016*), and a study in rats performing an effort-based decision-making task found that ovariectomy increased rats' choice of large, high-effort over small, low-effort rewards relative to their pre-surgery performance (*Uban et al., 2012*). To determine if sex differences in impulsive choice are influenced by gonadal hormones, males and females underwent either gonadectomy or sham surgeries. Consistent with the absence of estrous cycle effects on choice behavior, ovariectomies did not alter task performance, suggesting that if ovarian hormones contribute to the greater impulsive choice observed in females, their effects may be organizational rather than activational. Alternatively, local androgen synthesis in the brain (*Tobiansky et al., 2018*) might compensate for loss of circulating ovarian hormones to allow maintenance of pre-surgical patterns of choice behavior in females.

In contrast to the null effect of ovariectomies, orchiectomies increased impulsive choice relative to pre-surgery performance. Analyses of response latencies and trial omissions on forced-choice trials revealed patterns of performance in orchiectomized rats more akin to those in female rats than intact males (i.e., longer response latencies and more trial omissions at longer delays), suggesting that orchiectomy causes a shift toward reduced tolerance for delayed rewards. Importantly, these

results are not explained simply by motivational deficits, as orchiectomies did not increase omissions on free-choice trials. These findings are consistent with the results of a prior study in which a supra-physiological dose of testosterone administered to intact male rats caused a decrease in impulsive choice (*Wood et al., 2013*). Hence, testicular hormone signaling may allow bidirectional modulation of impulsive choice in adult rats.

The fact that orchiectomy shifted impulsive choice implies that testicular hormones do not solely contribute to the male-typical pattern of impulsive choice in an organizational manner, but that they play an activational role as well. Impulsive choice is mediated by a network of corticolimbic-basal ganglia structures, including prefrontal cortex, basolateral amygdala, and nucleus accumbens, as well as dopaminergic and serotonergic innervation of these structures (see *Floresco et al., 2008*; *Peters and Büchel, 2011*; *Bailey et al., 2016*; *Fobbs and Mizumori, 2017*; *Frost and McNaughton, 2017* for review). Androgen receptors are found within all of these brain regions, and their functions can be modulated by manipulations of testicular hormones (*Tobiansky et al., 2018*). Hence, it is likely that the effects of orchiectomy on impulsive choice are mediated at least in part via hormonal effects on these brain systems. Future studies involving manipulations of hormonal signaling within specific brain regions will be needed to address this issue.

## Limitations and conclusions

The results in the current study highlight two major findings. First, there was a significant and robust difference in impulsive choice between males and females that was not readily attributable to differences in food motivation or reward sensitivity. Second, we describe the novel finding that testicular hormones are necessary for maintaining the bias toward delayed gratification typical of males, whereas ovarian hormones do not appear to be necessary for maintaining the female-typical bias toward immediate gratification. Finally, the robust differences between males and females, as well as the differential roles of gonadal hormones, highlight the importance of assessing both sexes in studies of cost-benefit decision making (see *van den Bos et al., 2013*; *Orsini and Setlow, 2017* for reviews).

The current study had some of the largest group sizes to date in an assessment of sex differences in impulsive choice; however, it is limited in that rats of only a single strain and age were evaluated on a single task design using a fixed set of delays. As such, although the use of rodents avoids factors that can bias studies of impulsive choice in human subjects (such as age, education, and socio-economic status), it is not clear to what extent the current results are representative of human behavior. Indeed, a meta-analysis of studies assessing impulsive choice in human subjects found no sex difference, although the heterogeneity in subject ages and task designs renders it challenging to draw generalities (*Cross et al., 2011*). Particularly given that (in both rodents and humans) females appear to be more sensitive than males to punishment (*Cross et al., 2011*; *Orsini and Setlow, 2017*), it may be the case that greater impulsive choice among females will be most evident in circumstances involving actual (experienced, rather than hypothetical) rewards and delays.

Given the strong associations between extremes in impulsive choice and sex-prevalent psychiatric disorders (e.g., ADHD and substance use disorders associated with greater impulsive choice in males, and anorexia nervosa associated with reduced impulsive choice in females), it is interesting to speculate as to how such disorders might be related to sex differences in impulsive choice. Notably, the sex differences observed here (greater impulsive choice in females than males) are opposite of those predicted by the sex prevalence of psychiatric disorders associated with alterations in impulsive choice such as ADHD, SUDs, and anorexia nervosa. The results of the current study depict sex-atypical patterns of impulsive choice relative to sex differences in psychiatric disorders with which they are associated. In other words (and if the present findings in rats were extended to humans) male-prevalent psychiatric disorders are associated with male-atypical patterns of impulsive choice, whereas female-prevalent psychiatric disorders are associated with female-atypical patterns of impulsive choice. Although it is not clear whether and how such relationships are relevant to the etiology or symptoms of psychiatric disorders, future studies employing more targeted manipulations of gonadal hormone signaling may begin to address these questions.

# Materials and methods

## Subjects

Adult male (n = 16) and female (n = 16) Fischer 344 × Brown Norway F1 hybrid (FBN) rats were obtained at 6 months of age from the National Institute on Aging colony (Charles River Laboratories) and individually housed in the Association for Assessment and Accreditation of Laboratory Animal Care International-accredited vivarium facility in the McKnight Brain Institute building at the University of Florida. This study was performed in strict accordance with the recommendations in the Guide for the Care and Use of Laboratory Animals of the National Institutes of Health. All animal procedures were conducted according to protocols approved by the University of Florida Institutional Animal Care and Use Committee (protocol # 201904961). The facility was maintained at a consistent temperature of 25° with a 12 hr light/dark cycle (lights on at 0700) and free access to food and water except as otherwise noted. Rats were acclimated in this facility and handled for at least one week prior to initiation of any procedures. Group sizes were based on previous work from our labs that has assessed sex differences in decision making and PFC-dependent cognition (*Orsini et al., 2016*; *Blaes et al., 2019*). In this work, n = 8 rats/sex was determined to be needed to detect sex differences. Group sizes of n = 16/sex were employed to have sufficient rats of each sex for both GDX and sham conditions. Rats were tested in three separate cohorts, each separated by 6–12 months.

## Behavioral testing procedures

### Apparatus

Testing was conducted in eight identical standard rat behavioral test chambers (Coulbourn Instruments) with metal front and back walls, transparent Plexiglas side walls, and a floor composed of steel rods (0.4 cm in diameter) spaced 1.1 cm apart. Each test chamber was housed in a sound-attenuating cubicle and was equipped with a recessed food pellet delivery trough located 2 cm above the floor in the center of the front wall. The trough was fitted with a photobeam to detect head entries and a 1.12 W lamp for illumination. Food rewards consisted of 45 mg soy-free food pellets (5TUL, 1811155; Test Diet, Richmond, IN, USA). Two retractable levers were positioned to the left and right of the food trough (11 cm above the floor). An additional 1.12 W house light was mounted near the top of the rear wall of the sound-attenuating cubicle. A computer interfaced with the behavioral test chambers and equipped with Graphic State 4 software (Coulbourn Instruments) was used to control experiments and collect data.

### Behavioral shaping

Prior to shaping, rats were food restricted to 85% of their ad libitum weight. The intertemporal choice task was based on a design by *Evenden and Ryan, 1996* and was used previously to demonstrate age-related alterations in decision making in both Fischer 344 (*Simon et al., 2010*) and FBN (*Hernandez et al., 2017*) rats. Rats were initially shaped to lever press to initiate delivery of a food pellet into the food trough and were then trained to nosepoke in the food trough to initiate lever extension. Each nosepoke triggered extension of either the left or right lever (randomized across pairs of trials), a press on which yielded a single food pellet. After two consecutive days of reaching criterion performance (45 presses on each lever within 60 min), rats began testing on the intertemporal choice task.

### Intertemporal choice task

Each 80 min session consisted of 5 blocks of 12 trials each. Each 80 s trial began with a 10 s illumination of the food trough and house lights. A nosepoke into the food trough during this time extinguished the food trough light and triggered extension of either a single lever (forced-choice trials) or both levers simultaneously (free-choice trials). Each block began with two forced-choice trials (one for each lever) followed by 10 free-choice trials. The forced-choice trials were designed to remind rats of the delay contingencies in effect for that block. A press on one lever (either left or right, counterbalanced across age groups) resulted in one food pellet (the small reward) delivered immediately. A press on the other lever resulted in four food pellets (the large reward) delivered after a variable delay. The identities of the levers remained consistent throughout testing. Failure to press either lever within 10 s of its extension resulted in the levers being retracted and lights extinguished,

and the trial was scored as an omission. Once either lever was pressed, both levers were retracted for the remainder of the trial. The duration of the delay preceding large reward delivery increased between each block of trials (0, 10, 20, 40, 60 s; or 0, 2, 4, 8, 16 for females after ovariectomies), but remained constant within each block. See *Figure 1B* for task schematic.

### Estrous cycle tracking

After stable behavioral performance was achieved, vaginal lavages were conducted in females. Briefly, after daily behavioral testing, vaginal lavage samples were smeared on electrostatic glass slides and observed under a compound light microscope for cycle verification as in our previous work (*Orsini et al., 2016*). Each of the four phases of the estrous cycle was identified using specific criteria (*Marcondes et al., 2002*). For experimental consistency, male rats were handled in a manner similar to the females.

## Surgical procedures

### Ovariectomy

Females underwent either ovariectomy (OVX) or sham surgeries following stable baseline performance in the intertemporal choice task. Rats were anesthetized with isoflurane gas (1–5% in $O_2$) and were administered meloxicam (1 mg/kg), buprenorphine (0.05 mg/kg) and sterile saline (10 mL) subcutaneously. Rats were placed on their ventral side and the incision area on the back was swabbed with chlorhexidine. A 7 mm incision was then made through the outer skin and the underlying fascia was separated with hemostats. An additional 5 mm incision was made through the latissimus dorsi muscle to allow access to the peritoneal cavity. After clearing away adipose tissue, the uterine horns and ovaries were located, and the ovaries were retracted from the peritoneal cavity with forceps. The tissue between the ovaries and uterus was clamped with hemostats and ligated with absorbable sutures. Both ovaries were removed, and the uterus and associated tissue were placed back into the peritoneal cavity. The incision in the muscle was closed with absorbable sutures and the skin incision was closed with surgical clips. For sham surgeries, incisions were made through both the skin and muscle and closed with surgical clips and sutures, respectively. Immediately after surgery, rats received subcutaneous injections of warm saline. Buprenorphine was also administered 24 hr post-operation, and meloxicam 48–72 hr post-operation. A topical ointment was applied to facilitate wound healing, and all rats were placed on a heating pad for recovery. Clips were removed after 10–14 days and rats recovered for at least 2 weeks before food restriction and behavioral testing were reintroduced. Vaginal lavages were then conducted in OVX females for 7 days to confirm termination of estrous cycling, which manifested as a preponderance of leukocytes (as is seen in the diestrus phase in an intact female; *Montes and Luque, 1988*; *Larson and Carroll, 2007*). Sham females were handled in a similar manner across the same 7 days.

### Orchiectomies

Males underwent either orchiectomy or sham surgeries once they reached stable baseline performance. Rats were anesthetized with isoflurane gas (1–5% in $O_2$) and were administered meloxicam (1 mg/kg), buprenorphine (0.05 mg/kg) and sterile saline (10 mL) subcutaneously. Rats were placed on their ventral side and after disinfecting the area with chlorhexdine, a small incision was made at the tip of the scrotum and each testicle was separated. Another small incision was made in the scrotal sac to express the testicle with the tunica intact. The connective tissue was gently teased away and then the testis and epididymis were isolated. The vascular tissue of the spermatic cord was clamped, and the cord was then ligated 0.5 cm distal from the testis. The testis were then severed from the cord. These procedures were repeated for the second testicle. Once both testes were removed, the skin incision was sutured closed. For males receiving sham surgeries, an incision was made at the tip of the scrotum and then closed with sutures. Immediately after surgery, rats received subcutaneous injections of warm saline. Buprenorphine was also administered 24 hr post-operation, and meloxicam 48–72 hr post-operation. A topical ointment was applied to facilitate wound healing, and all rats were placed on a heating pad for recovery. Sutures were removed after 10–14 days and rats recovered for at least 2 weeks before food restriction and behavioral testing were reintroduced.

Two weeks after orchiectomy surgeries, blood was collected via tail nicks from sham and orchiectomized rats to determine serum testosterone levels using a testosterone ELISA kit (Abcam 108666).

Briefly, blood was allowed to clot at room temperature for a minimum of 30 min, and samples were then centrifuged at 4,500 RPM for 15 min at 4℃. The supernatant was transferred to a clean micro-centrifuge tube and stored at −80℃ until usage. All other steps were performed as per the manufacturer's suggestions except that sample serum volume was doubled (from 25 μL to 50 μL) to place serum testosterone levels within the level of detection. Sample dilution factors were accounted for in the analyses.

## Experimental design and statistical analyses

### Evaluation of sex differences in intertemporal choice

For experimental timeline, see *Figure 1A*. Raw data files were extracted using a Graphic State 4.0 analysis template that was custom-designed to extract the number of presses on each lever (large or small reward) during forced-choice and free-choice trials in each block of trials. The percentage of free-choice trials in each delay block on which rats chose the large, delayed reward was used as the primary measure of performance. Rats were trained for daily 15 sessions, after which performance was compared daily across a sliding five-session window using a two-factor, repeated measures ANOVA (delay × session) as in our previous work (*Simon et al., 2007*; *Simon et al., 2010*; *Hernandez et al., 2017*). Stability was defined as the absence of both a main effect of session and a session × delay interaction. Data from the first stable 5-session window were averaged to calculate a mean percent choice of the large reward at each delay (0, 10, 20, 40, 60 s). Sex comparisons were conducted using a two-factor, repeated measures ANOVA, with sex as a between-subjects factor and delay as a within-subjects factor. Lever response latencies and total trials completed were also compared between sexes. Response latency data (the time between lever extension and a lever press) were collected on forced-choice trials. Previous work shows that response latencies can differ for large and small reward levers (*Hernandez et al., 2017*) and may reflect differences in motivation to obtain differently-preferred rewards (*Crespi, 1942*; *Setlow et al., 2003*; *Mai et al., 2012*; *Shimp et al., 2015*). Sex comparisons of latency data were conducted using a three-factor, repeated measures ANOVA, with sex as a between-subjects factor and both lever type (small and large) and delay as within-subjects factors. Note that only completed trials were included in the analyses of response latencies (i.e., omitted trials were not included). Forced-choice trial omissions were analyzed using the same repeated measures ANOVA as the latency analyses. Either independent-sample or paired-sample t-tests were used to determine differences (either group or lever) within a delay. Finally, the percentages of free-choice trial omissions were compared using an independent samples t-test. All analyses were conducted in SPSS 26, and the alpha was set to 0.05.

### Evaluation of food motivation in intertemporal choice

To determine if sex differences in choice performance were due to differences in motivation to obtain food under food restriction conditions, rats were tested in the intertemporal choice task following both a 1 hr and a 24 hr ad libitum feeding schedule. To counterbalance testing, half of the males and females in the cohort were placed on a 1 hr ad libitum feeding schedule prior to testing on the intertemporal choice task on the first day. On the following day, all rats were tested under the usual food-restricted conditions (as a 'washout period' for the 1 hr ad libitum feeding schedule). On the third day, the second half of the cohort was given the 1 hr ad libitum feeding schedule. After a 48 hr washout period, rats were given a 24 hr ad libitum feeding schedule using the same design as the 1 hr feeding schedule. Comparisons of choice performance under the two ad libitum feeding conditions were conducted using a three-factor, repeated measures ANOVA, with sex as a between-subjects factor and both feeding condition (ad libitum vs. food-restricted) and delay as within-subjects factors. The percentages of free-choice trial omissions were compared using a two-factor, repeated measures ANOVA with sex as a between-subjects factor and feeding condition (ad libitum vs. food restricted) as a within-subjects factor.

### Evaluation of gonadectomy in intertemporal choice

Procedures used to analyze data after surgery were identical to those used to analyze data prior to surgery. Data from a stable five session window were averaged to calculate a mean percent choice of the large reward at each delay (0, 10, 20, 40, 60 s). Group comparisons were conducted in males and females separately using a three-factor, repeated measures ANOVA, with surgery group (sham

vs. gonadectomy) as a between-subjects factor and surgery phase (pre- vs. post-surgery) and delay as within-subjects factors. Comparisons of forced-choice lever response latency and forced-choice trial omission data were conducted within each surgical group using a three-factor, repeated measures ANOVA, with surgery phase (pre- vs. post-surgery), lever type (small vs. large), and delay as within-subjects factors. Finally, the percentages of free-choice trial omissions were compared using a two-factor, repeated measures ANOVA with surgery group (sham vs. gonadectomy) as a between-subjects factor and surgery phase (pre- vs. post-surgery) as a within-subjects factor.

## Acknowledgements

We thank Vicky S Kelly, Matthew M Bruner, Shannon C Wall, and Bonnie I McLaurin for technical assistance. Supported by R01AG029421 and the McKnight Brain Research Foundation (JLB), RF1AG060778 (JLB, BS, CJF), a McKnight Predoctoral Fellowship and the Pat Tillman Foundation (CMH), and a Thomas H Maren Fellowship and K99DA041493 (CAO).

## Additional information

### Funding

| Funder | Grant reference number | Author |
|---|---|---|
| National Institutes of Health | R01AG029421 | Jennifer L Bizon |
| National Institutes of Health | RF1AG060778 | Barry Setlow Jennifer L Bizon |
| National Institutes of Health | K99DA041493 | Caitlin Orsini |
| McKnight Brain Research Foundation | | Jennifer L Bizon |
| McKnight Foundation | | Caesar M Hernandez |
| Pat Tillman Foundation | | Caesar M Hernandez |
| Thomas H. Maren Foundation | | Caitlin Orsini |

The funders had no role in study design, data collection and interpretation, or the decision to submit the work for publication.

### Author contributions

Caesar M Hernandez, Conceptualization, Data curation, Formal analysis, Supervision, Investigation, Writing - original draft, Project administration, Writing - review and editing; Caitlin Orsini, Investigation, Methodology, Writing - review and editing; Alexa-Rae Wheeler, Conceptualization, Formal analysis, Supervision, Investigation, Writing - review and editing; Tyler W Ten Eyck, Sara M Betzhold, Noelle G Wright, Formal analysis, Investigation, Writing - review and editing; Chase C Labiste, Formal analysis, Methodology, Writing - review and editing; Barry Setlow, Supervision, Funding acquisition, Project administration, Writing - review and editing; Jennifer L Bizon, Conceptualization, Supervision, Funding acquisition, Project administration, Writing - review and editing

### Author ORCIDs

Caesar M Hernandez https://orcid.org/0000-0001-9690-5119
Caitlin Orsini https://orcid.org/0000-0001-5644-2316
Barry Setlow https://orcid.org/0000-0001-9133-9445
Jennifer L Bizon https://orcid.org/0000-0002-9517-5844

### Ethics

Animal experimentation: This study was performed in strict accordance with the recommendations in the Guide for the Care and Use of Laboratory Animals of the National Institutes of Health. All animal procedures were conducted according to protocols approved by the University of Florida Institutional Animal Care and Use Committee (protocol # 201904961).

Decision letter and Author response
Decision letter https://doi.org/10.7554/eLife.58604.sa1
Author response https://doi.org/10.7554/eLife.58604.sa2

## Additional files

### Supplementary files
• Transparent reporting form

### Data availability
All data generated during this study are included in the manuscript and supporting files.

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
