## [Decision Letter]

**Acceptance summary:**

The reported findings suggest that sex differences in impulsive responding in rats are due to testicular hormones. This research make a significant contribution to the field, enhancing our understanding of sex differences in impulsivity.

**Decision letter after peer review:**

Thank you for submitting your article "Testicular hormones mediate robust sex differences in impulsive choice" for consideration by *eLife*. Your article has been reviewed by three peer reviewers, including Shelly B Flagel as the Reviewing Editor and Reviewer #1, and the evaluation has been overseen by Kate Wassum as the Senior Editor.

The reviewers have discussed the reviews with one another and the Reviewing Editor has drafted this decision to help you prepare a revised submission.

Summary:

The authors investigate the impact of sex and gonadal hormones on impulsive responding in rats. While this topic is not novel, the results in the literature are mixed, and the authors attempted to dissociate sex and hormonal factors using a strong within-subject design. They report sex differences, with females showing more impulsive responding relative to males and conclude that this difference is likely due to testicular hormones, as gonadectomy affected male, but not female, responding. These gonadectomy results were considered to be a valuable contribution to the understanding of the influence of sex on impulsivity.

The reviewers agreed that this manuscript is well-written, that the data are easy to assimilate and beautifully graphed, and that the concluding paragraph in particular gives much food for thought. However, they feel that the conclusions need to be tempered and that further analyses and discussion are warranted surrounding some of the points highlighted below.

Essential revisions:

1) One primary concern with this manuscript is that it is aiming to resolve a big question that others have tried to resolve before this group, namely whether there are sex differences in performance of delay discounting tasks. The current data clearly imply that there are, and that females are more impulsive, but previous groups were likely just as confident in their assessment. Why should this study be taken as being more definitive than any of the others that preceded it? What was done better or differently than others? Is this strain of rats more likely to represent the human condition, or is the behavioral methodology better for comparative psychology? The strengths and weakness of the current approach should be made explicit in the Discussion.

2) To be clear, the above questions do not invalidate the scientific study done here- in fact, review of the studies cited suggests that this is the largest study done to date (n = 16 per group vs. n = 4 (van Haaren)/6 (Eubig)/9-10 (Sackett)etc.) but it is still only a small number of animals. Although this is a typical sample size for an animal study, due to the greater experimental control inherent when using animal models, it is still possible to get results that are not indicative of the whole population, especially as commercial rodent suppliers generally tend to ship siblings rather than genetically diverse cohorts. Likewise, individual experiments using human subjects may find evidence for or against sex/gender differences due to a relatively small n. However, a meta-analysis found no gender differences in delay discounting tasks across 277 studies using human subjects (Cross et al., 2011).

3) Beyond the issue of sample size, there are also strain differences in delay discounting tasks (e.g. (Garcia and Kirkpatrick, 2012; Huskinson et al., 2012)) So, again, if this study is aiming to definitively demonstrate sex difference in delay discounting, large numbers of animals should be used and the effect should hold across multiple strains. I therefore urge the authors to temper their title and conclusions: although compelling, the current data are unlikely to resolve the issue of whether sex differences are present in the performance of delay-discounting tasks, but will simply be added in to the collection of studies, adding weight to one side of the argument, but certainly not ending it.

4) The repeated testing warrants further discussion. In relation, it seems that comparing males and females (with and without surgery) would be important in order to make claims about whether or not hormones are truly mediating the sex differences. Given the current analysis, it is difficult to say if gonadectomy rendered males more similar to females.

5) It is not clear that the described differences are not due to overall differences in motivation or reward magnitude. While the data suggest that males and females respond similarly to restricted feeding, it doesn't directly address the motivational state of females at the time of the original test. Further, the sex differences in latency and omitted trials highlights potential differences in motivational states. Additional analyses might be telling in this regard. For example, if only non-omitted trials are included, are there still sex differences in latency? Further, in their restricted feeding tests, more females than males are excluded due to omitted trials and this could be a meaningful sex difference. This warrants further discussion.

6) In relation to the point above, the food restriction data are useful, but suffer from a small effect size. It is hard to see how the females could become any more impulsive, and the lines seem to be on top of one another, yet there is no main effect of sex in the omnibus ANOVA. Given that these results are integral in proving that the effect in females is not dependent on differences in satiety for sugar pellets, it would be useful to analyse the data from females only and show there is a clear effect of food restriction in these rats.

7) Further, it seems that the longer response latencies and greater percentage of trial omissions seen in females on the forced choice trials for the larger reward could also indicate that this large reward isn't as rewarding for females as males? This is partially addressed by the 24-hr ad lib feeding study but not entirely as ad lib feeding seems to affect females differently than males (note the decrease in % choice at the 0 sec delay and the increase in omissions seen in females after 24hr ad lib feeding).

8) Post-hoc comparisons should be included in the primary text when appropriate. For example, it is stated, "As shown in Figure 2A, females made fewer choices of large, delayed rewards than males (two-factor ANOVA, main effect of sex: F_(1,30)_=10.586, p=0.003), particularly when longer delays preceded delivery of the large reward (sex x delay interaction, F_(4,120)_=8.572; p<0.001). However, the graph suggests that sex differences were more robust at shorter delays. Post-hoc comparisons should be included for each delay.

9) In the Introduction, the authors mention previous work showing that testosterone administration in rats can influence impulsive choice whereas work in humans does not recapitulate these effects. The authors should discuss how this incongruity in the literature may influence the translational interpretation of their findings.

---

## [Author Response]

Essential revisions:1) One primary concern with this manuscript is that it is aiming to resolve a big question that others have tried to resolve before this group, namely whether there are sex differences in performance of delay discounting tasks. The current data clearly imply that there are, and that females are more impulsive, but previous groups were likely just as confident in their assessment. Why should this study be taken as being more definitive than any of the others that preceded it? What was done better or differently than others? Is this strain of rats more likely to represent the human condition, or is the behavioral methodology better for comparative psychology? The strengths and weakness of the current approach should be made explicit in the Discussion.

We thank the reviewer for their comments. We agree that other groups were likely just as confident in their findings, but we believe that our data provide the most convincing evidence to date for greater impulsive choice in female compared to male rats. Prior rodent studies in agreement with our findings either had small sample sizes or were not designed to explicitly test sex differences. One study employing an intertemporal choice task in which delays for the large reward were varied across sessions in 4 male and 4 female Wistar rats found that females discounted large food rewards to a greater extent than males when the delay to delivery increased from 9 to 36 s (van Haaren et al., 1988). Similarly, another study using rats bred for variability in saccharin preference employed an adjusting delays task and found that female rats were more impulsive relative to males (Perry et al., 2007). This sex difference was conditional, however, in that it was only evident in rats with low preference for saccharin (but not high preference for saccharine).

Two more recent studies also reported evidence for numerically greater impulsive choice in female than male rats, although the differences did not reach statistical significance (Eubig et al., 2014; Lukkes et al., 2016). In the first study, Eubig et al. employed an intertemporal choice task design similar to that used in our study and found that female rats had numerically fewer choices of large, delayed rewards relative to males. In the second study, Lukkes et al. also employed an intertemporal choice task design similar to that used in our study to evaluate impulsive choice in adolescent and adult male and female rats. These authors reported a numerically lower indifference point (greater impulsive choice) in female relative to male rats. It is possible that neither of these studies were sufficiently powered to detect statistically significant sex differences, given their smaller group sizes (Eubig et al.: n=6/sex; Lukkes et al.: n=8-9/sex) compared to our study (n=16/sex).

One study using mice in an intertemporal choice task in which mice had to nosepoke into food troughs that delivered a small immediate food reward vs. a large, delayed food reward reported that females were more impulsive than males (Koot et al., 2009). Like the Perry et al., 2007 study described above, however, these results were conditional in that the sex difference was only present in “high-discounting” mice, whereas there were no sex differences in “low-discounting” mice.

In contrast to the (limited) evidence for greater impulsive choice in female relative to male rodents described above, two studies reported neither numerical nor statistically significant sex differences in impulsive choice (Perry et al., 2008; Sackett et al., 2019). Although Perry and colleagues previously reported greater impulsive choice in females relative to males (Perry et al., 2007), these authors reported no such difference in a subsequent study (Perry et al., 2008). The null effect in the 2008 study could have been due to strain differences, as their previously reported sex differences (Perry et al., 2007) were in a subset of Sprague-Dawley rats bred for saccharin preference, whereas the lack of differences in the 2008 paper was in Wistar rats. The Sackett et al., 2019 study tested Long-Evans rats in an intertemporal choice task similar to that used in our study and found no differences between males and females in their choice behavior. Despite a null effect of sex in this study, however, when the authors used rats’ individual variability in choice behavior to sort them into high discounters (i.e., more impulsive choice) and low discounters (i.e., less impulsive choice), the high discounter group contained double the number of females relative to males, whereas the low discounting group contained double the males relative to females.

The results of the current study explicitly and unequivocally show a greater preference for small, immediate rewards in females relative to males and are in agreement (at least numerically if not statistically) with the bulk of the rodent literature on the topic, across rodent species and strains. Although the strain used in our study could certainly account for the large sex difference (relative to most previous work), we suspect that the larger sample sizes and the use of longer delays to the large reward delivery relative to most previous studies played a greater role in our ability to detect sex differences. Note that we cannot claim that the Fischer 344 × Brown Norway F1 hybrid rats used in our study is more representative of humans than other strains.

We now provide a more expanded discussion of these issues in the Discussion section.

2) To be clear, the above questions do not invalidate the scientific study done here- in fact, review of the studies cited suggests that this is the largest study done to date (n = 16 per group vs. n = 4 (van Haaren)/6 (Eubig)/9-10 (Sackett)etc.) but it is still only a small number of animals. Although this is a typical sample size for an animal study, due to the greater experimental control inherent when using animal models, it is still possible to get results that are not indicative of the whole population, especially as commercial rodent suppliers generally tend to ship siblings rather than genetically diverse cohorts. Likewise, individual experiments using human subjects may find evidence for or against sex/gender differences due to a relatively small n. However, a meta-analysis found no gender differences in delay discounting tasks across 277 studies using human subjects (Cross et al., 2011).

The reviewer emphasizes an important point. Ideally, we would reproduce these results across multiple strains using identical methodology to support our claim; however, it is important to note that among rodent studies (including several using different rat strains), the preponderance of evidence seems to point to greater impulsive choice in females (see response to Point 1 above). Notably, we observed considerable individual variability among male rats this variability was less evident in females, likely due to a floor effect; (see Figure 4B and E). It will be important in future work to follow up on these individual differences, as there is considerable evidence (largely from male rats) that variability in impulsive choice predicts numerous other behavioral phenotypes relevant to neuropsychiatric disorders. One additional important point to make is that the rats in this study were run in 3 separate cohorts each separated by 12-24 months. As such, our detection of sex differences cannot be solely attributed to the use of rats from only a few different litters (i.e., reduced genetic variability). This latter point is now noted in the subsection “Subjects”.

Regarding the null findings of the meta-analysis done by Cross et al., 2011, we agree it could call into question the broader translational relevance of our findings, but we also believe that gender differences in impulsive choice may be conditional and nuanced. There are several studies in the human literature to suggest sex differences consistent with our findings (e.g., Logue and Anderson, 2001, Reynolds et al., 2006, Smith and Hantula, 2008, Beck and Triplett, 2009); however, factors such as the use of real vs. hypothetical rewards may influence the magnitude and direction of gender differences in impulsive choice (Kirby and Maraković, 1995; Kirby and Maraković, 1996; Weafer and de Wit, 2014). Considered together, it is possible that inconsistencies in gender/sex differences in impulsive choice can be attributed to the types of rewards used or the weights placed on distinct features of rewards. Particularly if delays are viewed as more “costly” or “punishing” by females (who tend to be more sensitive to punishments than males; Cross et al., 2011), it may be the case that greater impulsive choice among females will be most evident in circumstances that more closely model actual (experienced, rather than hypothetical) delays. We now mention this point in the subsection “Limitations and Conclusions”.

3) Beyond the issue of sample size, there are also strain differences in delay discounting tasks (e.g. (Garcia and Kirkpatrick, 2012; Huskinson et al., 2012)) So, again, if this study is aiming to definitively demonstrate sex difference in delay discounting, large numbers of animals should be used and the effect should hold across multiple strains. I therefore urge the authors to temper their title and conclusions: although compelling, the current data are unlikely to resolve the issue of whether sex differences are present in the performance of delay-discounting tasks, but will simply be added in to the collection of studies, adding weight to one side of the argument, but certainly not ending it.

We thank the reviewer for this comment, and again we agree. We have now changed the title to clarify that the study was conducted in rats and have incorporated a more detailed discussion of the rodent literature within the manuscript (Discussion). We believe that our data are in agreement with the majority of prior evidence for greater impulsive choice in female rodents (see responses above). Indeed, across strains, to our knowledge, there are more studies showing greater impulsive choice in female rodents than not, although we concede the limitations of the use of a single rodent strain and task design in our study (subsection “Limitations and Conclusions”).

4) The repeated testing warrants further discussion. In relation, it seems that comparing males and females (with and without surgery) would be important in order to make claims about whether or not hormones are truly mediating the sex differences. Given the current analysis, it is difficult to say if gonadectomy rendered males more similar to females.

Repeated testing of the same animals before and after surgery allowed us to incorporate both within- and between-subjects comparisons of the effects of gonadectomy (GDX). In our view, this was the strongest possible experimental design as it allowed us to include both surgical condition and timepoint as variables (which in turn enabled us to determine the contributions of GDX vs. simply the passage of time from pre- to post-surgery). The reviewer makes a good point regarding claims about whether or not hormones mediate the sex difference. Although we would argue that the results in castrated males are certainly suggestive, additional experiments would be necessary to strongly draw this conclusion, and thus we have moderated our language on this point in the revised manuscript (Discussion).

5) It is not clear that the described differences are not due to overall differences in motivation or reward magnitude. While the data suggest that males and females respond similarly to restricted feeding, it doesn't directly address the motivational state of females at the time of the original test. Further, the sex differences in latency and omitted trials highlights potential differences in motivational states. Additional analyses might be telling in this regard. For example, if only non-omitted trials are included, are there still sex differences in latency? Further, in their restricted feeding tests, more females than males are excluded due to omitted trials and this could be a meaningful sex difference. This warrants further discussion.

The reviewer brings up an important point regarding the potential contributions of sex differences in motivation, and we have attempted to clarify this issue throughout the Results section. In particular, we did not distinguish well in our original manuscript between omissions on free- vs. forced-choice trials, which we have attempted to do throughout the revised manuscript.

We now clarify (subsection “Effect of sex on intertemporal choice performance”) that there was no sex difference in the number of omitted free-choice trials during the original comparison of performance under food restricted conditions (i.e., for the data shown in Figure 2). As shown by the data in Figure 4C and F, pre-feeding (which should reduce motivation to obtain food during subsequent task performance) causes an increase in free-choice trial omissions, suggesting that the number of free-choice trial omissions is a reasonable proxy for food motivation (see Simon et al. 2009, Neuropsychopharmacology for similar findings). Note that there were also no sex differences in omitted free-choice trials during these pre-feeding tests as well (subsection “Effect of food motivational state on intertemporal choice performance in males and females”). Given these data (as well as the absence of sex differences in choice behavior in the effects of varying degrees of pre-feeding), we believe that the data argue for no sex differences in primary motivation to obtain food, at least under the conditions required for intertemporal choice task performance.

Regarding the sex differences in response latencies and trial omissions on forced-choice trials (Figure 3), we agree with the reviewer that these data highlight sex differences in motivation; however, we argue that these data are more consistent with sex differences in motivation specifically to obtain the large, delayed reward rather than differences in food motivation in general. In females, the increases in latency and omitted trials were largely limited to choices of the large reward. Indeed, latencies to obtain the small reward did not change at all across blocks of trials in females (Figure 3A), and on neither measure did small reward latencies or small reward trial omissions differ by sex. If females were less food motivated than males (either in general or as a function of satiation as they progressed through the blocks of trials), we would expect this to be reflected in longer latencies and more trial omissions to an equivalent degree on both small and large reward trials. As the changes in latency and trial omissions in females were mostly selective to the large reward, we conclude that this was due to greater impulsive choice/delay intolerance in females. This argument is now presented in greater detail in the subsection “Effect of sex on intertemporal choice performance”.

Regarding the latency measures, we now clarify that only completed trials were included in the analyses of latency data (i.e., omitted trials were excluded from this analysis – this is now clarified in the subsection “Evaluation of sex differences in intertemporal choice”). The reviewer makes a good point regarding the greater number of excluded females than males in the pre-feeding test, however, and we now mention this point in the subsection “Greater impulsive choice in females is not dependent on differential hunger/satiety”.

6) In relation to the point above, the food restriction data are useful, but suffer from a small effect size. It is hard to see how the females could become any more impulsive, and the lines seem to be on top of one another, yet there is no main effect of sex in the omnibus ANOVA. Given that these results are integral in proving that the effect in females is not dependent on differences in satiety for sugar pellets, it would be useful to analyse the data from females only and show there is a clear effect of food restriction in these rats.

We thank the reviewer for catching our errors in statistical reporting! In the revised manuscript we now clarify the results from the pre-feeding experiment. The 1-hr ad libitum feeding schedule caused n=2 male and n=5 female rats to omit enough choices to be excluded from the analysis of choice performance (leaving n=14 male and n=11 female) but not the analysis of free-choice trial omissions (for which all rats were included). For choice behavior, a multi-factor ANOVA (sex × feeding schedule × delay) confirmed the expected main effects of sex (F_(1,23)_=26.753, p<0.001), delay (F_(4,92)_=112.833, p<0.001) and the sex × delay interaction (F_(4,92)_=11.227, p<0.001; Figure 4A). The 1-hr ad libitum feeding schedule decreased choice of large, delayed rewards (main effect of schedule: F_(1,23)_=5.861, p=0.024; Figure 4B) in a non-delay-dependent manner (schedule × delay: F_(4,92)_=1.368, p=0.251). Most importantly, however (and most relevant to the point being tested in this experiment), there were no interactions between sex and the 1-hr ad libitum feeding schedule (schedule × sex: F_(1,23)_=0.417, p=0.525; schedule × sex × delay: F_(4,92)_=1.270, p=0.288), suggesting that different levels of hunger/satiation in males and females did not account for the sex difference in choice performance. Note that there was a main effect of feeding schedule (F_(1,30)_=9.115, p=0.005; Figure 4C) on the percentage of free-choice trials omitted such that 1-hr ad libitum significantly increased trial omissions irrespective of sex (main effect of sex: F_(1,30)_=0.294, p=0.592; schedule × sex: F_(1,30)_=0.294, p=0.592) – see also the response to Point 5 above.

The 24-hr ad libitum feeding schedule caused n=3 female rats to omit enough choices to be excluded from the choice but not the trial omission analysis. A multi-factor ANOVA (sex × feeding schedule × delay) confirmed the expected main effects of sex (F_(1,27)_=13.414, p=0.001) and delay (F_(4,108)_=81.398, p<0.001) and the sex × delay interaction (F_(4,108)_=7.899, p<0.001; Figure 4D). Unlike the 1-hr ad libitum schedule, there was no decrease in the choice of large, delayed rewards as a result of the 24-hr ad libitum feeding schedule (main effect of feeding schedule F_(1,27)_=2.302, p=0.141; Figure 4E). Most importantly (and as with the 1-hr schedule), however, there was no interaction between sex and schedule (schedule × sex: F_(1,27)_=0.021, p=0.887; schedule × sex × delay: F_(4,108)_=1.433, p=0.228), indicating that a sex difference in food motivation likely does not account for the sex difference in choice performance. Importantly, as with the 1-hr ad libitum schedule, there was a significant of effect of feeding schedule on trial omissions such that more trials were omitted during the 24-hr ad libitum period (main effect of schedule: F_(1,30)_=13.400, p=0.001; Figure 4F) irrespective of sex (main effect of sex: F_(1,30)_=2.297, p=0.140; schedule × sex: F_(1,30)_=2.839, p=0.102). We now include the corrected statistics for this experiment in the text (subsection “Effect of sex on intertemporal choice performance”).

7) Further, it seems that the longer response latencies and greater percentage of trial omissions seen in females on the forced choice trials for the larger reward could also indicate that this large reward isn't as rewarding for females as males? This is partially addressed by the 24-hr ad lib feeding study but not entirely as ad lib feeding seems to affect females differently than males (note the decrease in % choice at the 0 sec delay and the increase in omissions seen in females after 24hr ad lib feeding).

The reviewers bring up an excellent point. We have now highlighted the absence of difference between males and females in free-choice trial omissions under baseline conditions (subsection “Effect of sex on intertemporal choice performance”), suggesting a comparable willingness to engage in the task (see also the response to point 5 above). We also now discuss the rats’ performance at the 0 s delay (Figures 2 and 3). Specifically, the fact that males and females show equivalent preference for the large reward in the absence of delays (Figure 2A) suggests robust detection of reward magnitude differences and preference for the larger reward in females. At the 0 s delay, females did show slightly (though not significantly) longer latencies than males to choose the large reward (Figure 3A, t_(27)_ = -2.031, p=0.052; mean difference -0.222 s), but this sex difference was an order of magnitude smaller than the sex difference at longer delays such as 40 s (t_(27)_ = -5.453, p<0.001; mean difference -3.33 s) or 60 sec (t_(27)_ = -3.103, p=0.004; mean difference -2.26 s). In addition, there was no sex difference in the number of forced choice large reward trial omissions at 0 sec delays (Figure 3B, t_(30)_=1.192, p=0.243). Considered together, these data provide little to no evidence that females were less motivated by the large reward *per se*, but rather are consistent with the idea that any reductions in motivation for the large reward were due to the delays imposed before its delivery in later blocks of trials.

In addition, we compared males and females at the 0 sec delay in both restricted and ad libitum conditions and did not observe any significant differences under restricted (t_(23)_ = 1.888, p=0.072) or ad libitum (t_(23)_ = 1.401, p=0.175) conditions (Figure 4A). Furthermore, paired samples t-tests comparing restricted and ad libitum conditions at 0 sec within sexes revealed no significant difference in males (t_(13)_ = 1.0, p=0.336) or females (t_(10)_ = 1.019, p=0.332). Closer inspection of the data highlighted that n=1 female rat decreasing choice of the large reward at 0 sec from 90% to 60% was responsible for driving the numerical difference.

8) Post-hoc comparisons should be included in the primary text when appropriate. For example, it is stated, "As shown in Figure 2A, females made fewer choices of large, delayed rewards than males (two-factor ANOVA, main effect of sex: F_(1,30)_=10.586, p=0.003), particularly when longer delays preceded delivery of the large reward (sex x delay interaction, F_(4,120)_=8.572; p<0.001). However, the graph suggests that sex differences were more robust at shorter delays. Post-hoc comparisons should be included for each delay.

We thank the reviewer for their comment. We now indicate in each figure the delays at which the groups or conditions differed significantly.

9) In the Introduction, the authors mention previous work showing that testosterone administration in rats can influence impulsive choice whereas work in humans does not recapitulate these effects. The authors should discuss how this incongruity in the literature may influence the translational interpretation of their findings.

We thank the reviewer for their comment. The previous work in rats involved very high levels of testosterone administration (the intention in this study was to model anabolic steroid abuse). In humans, whereas exogenous testosterone (at doses an order of magnitude lower than those used in abuse settings) does not seem to modulate impulsive choice (Ortner et al., 2013), salivary testosterone levels do associate with discounting of delayed gains (Takahashi et al., 2006), suggesting that testosterone levels within the physiological range may play a role in impulsive choice. We have now clarified this point in the Introduction.